



# 1 Impacts of anthropogenic water regulation on global
# 2 riverine dissolved organic carbon transport

Yanbin You[1,2], Zhenghui Xie[1,2*], Binghao Jia[1], Yan Wang[3], Longhuan Wang[1], Ruichao
Li[1], Heng Yan[1,2], Yuhang Tian[1,2], Si Chen[1,2]
[1]State Key Laboratory of Numerical Modeling for Atmospheric Sciences and Geophysical Fluid
Dynamics, Institute of Atmospheric Physics, Chinese Academy of Sciences, Beijing 100029, China
[2]College of Earth and Planetary Sciences, University of Chinese Academy of Sciences, Beijing 100049,
China
[3]State Key Laboratory of Hydrology-Water Resources and Hydraulic Engineering, Nanjing Hydraulic
Research Institute, Nanjing 210029, China
*Correspondence to*: Zhenghui Xie (zxie@lasg.iap.ac.cn)
**Abstract.** Anthropogenic water regulation activities, including reservoir interception, surface water
withdrawal, and groundwater extraction, alter riverine hydrologic processes and affect dissolved organic
carbon (DOC) export from land to rivers and oceans. In this study, schemes describing soil DOC leaching,
riverine DOC transport, and anthropogenic water regulation were developed and incorporated into the
Community Land Model 5.0 (CLM 5.0) and the River Transport Model (RTM). Three simulations by the
developed model were conducted on a global scale from 1981 to 2013 to investigate the impacts of
anthropogenic water regulation on riverine DOC transport. The validation results showed that DOC
exports simulated by the developed model were in good agreement with global river observations. The
simulations showed that DOC transport in most rivers was mainly influenced by reservoir interception
and surface water withdrawal, especially in central North America and eastern China. Four major rivers,
including the Danube, Yangtze, Mississippi, and Ganges Rivers, have experienced reduced riverine DOC
flows due to intense water management, with the largest effect occurring in winter and early spring. In
the Danube and Yangtze River basins, the impact in 2013 was four to five times greater than in 1981,
with a retention efficiency of over 50 %. The Ob River basin was almost unaffected. The total impact of
anthropogenic water regulation reduced global annual riverine DOC exports to the ocean by
approximately 13.36 Tg C yr$^{-1}$, and this effect increased from 4.83 % to 6.20 % during 1981–2013,
particularly in the Pacific and Atlantic Oceans.




## 1. Introduction

Rivers are a pipe linking the two major carbon pools of terrestrial and ocean ecosystems and are one of
the key hubs of the global carbon cycle (Cole et al., 2007). According to the IPCC AR5, terrestrial
ecosystems deliver about 1.7 Pg C per year to rivers through surface and subsurface runoff and about 0.9
Pg C per year to oceans via rivers, of which about 0.21 Pg is dissolved organic carbon (DOC) (Ludwig
et al., 1996). This is equivalent to about 1 % of the global net primary productivity (NPP) of terrestrial
ecosystems (Zhang, 2012). Riverine DOC is a higher reactive organic carbon, is easily decomposed, and
is a direct source of carbon for microbial food webs in rivers and oceans, as well as a source of greenhouse
gas emissions from freshwater systems (Li et al., 2019; Tranvik & Jansson, 2002). It deeply affects the
biogeochemical cycles of rivers and offshore ecosystems. Therefore, it is important to clarify the
transport characteristics of riverine DOC for estimating global carbon budgets.
In recent years, anthropogenic water management activities, including reservoir interception, surface
water withdrawal, and groundwater extraction, have intensified the degree of interference with natural
processes on the surface of river basins, altered the hydrological and hydraulic processes of rivers, and
affected material circulation and transportation (Zhang, 2012). For example, extraction from rivers,
reservoirs, and underground aquifers affects hydrological systems, leading to a reduction in subsurface
runoff and eventually to decreased soil DOC leaching (Zeng et al., 2016), whereas activities such as
irrigation can lead to increased surface runoff, resulting in increased soil carbon losses (Ren et al., 2016).
Artificially constructed large reservoirs or dams disrupt the carbon cycle balance of the river continuum
in its natural state (Maavara et al., 2017), resulting in retention of DOC and sediment, while lower river
velocities and higher material concentrations lead to increased microbial activity in the water body, thus
changing the nutrient state of the river ecosystem (Liu et al., 2022). However, the impact of these
anthropogenic disturbances on riverine carbon transport has been ignored in estimating the global carbon
budget (Regnier et al., 2013).
Based on field surveys involving global riverine DOC transport flux estimation, the United Nations
Environment Programme has constructed a world river discharge database, GEMS-GLORI, that lists 48
attributes of 555 major world rivers (Meybeck, 1982; Meybeck & Ragu, 2012). There are also regional
survey programs, such as the Pan-Arctic River Transport of Nutrients, Organic Matter, and Suspended
Sediments (PARTNERS, https://arcticgreatrivers.org/) and the United States Geological Survey (USGS)



Data Center (https://waterdata.usgs.gov/nwis), which provide riverine organic carbon flux data for parts
of large rivers. Field survey studies are directly limited by data availability and completeness and
therefore mostly focus on large rivers in developed regions, making it difficult to cover rivers in other
regions. Moreover, only annual averages are usually available, with no time-series variation. Some
researchers have started to explore the mechanisms of riverine carbon flux changes using empirical
statistical models, which combine observed data with driving factors including river basin characteristics
(Ludwig et al., 1996), soil carbon and nitrogen ratios (Aitkenhead & McDowell, 2000), land-cover types
(Harrison et al., 2005), and river discharge (Fabre et al., 2020). However, the empirical statistical method
does not consider complex ecological processes within the watershed and cannot describe material
changes in the river network in detail. To identify changes in carbon transport and its driving mechanisms
spatially and explicitly, numerous process-based numerical models are currently used for DOC transport
simulations. Futter et al. (2007) proposed the integrated catchments model for carbon (INCA-C), which
explicitly considers land use, hydrological processes, soil carbon biogeochemical cycles, and surface
water processes. Liao et al. (2019) developed a three-dimensional terrestrial ecosystem model (ECO3D)
considering the influence of lateral water flows. These models simulate regional riverine DOC dynamics
more accurately than earlier models, but their accuracy relies on complex parametric schemes of eco-
hydrological processes and extensive data surveys, so that it is difficult to extend these models to global-
scale simulations. Wu et al. (2014) integrated ecological driving factors and biogeochemical processes
to develop a TRIPLEX-DOC model that predicts DOC metabolism, sorption, desorption, and loss
processes in soils. Li et al. (2019) added a river hydrological process module to construct the TRIPLEX-
HYDRA model and applied it to simulate global riverine DOC fluxes. However, the model did not
consider the impact of human activities on riverine DOC transport. Tian et al. (2015) constructed the
dynamic land ecosystem model (DLEM), a fully distributed model that integrates vegetation dynamics
with processes such as water, carbon, nitrogen, and phosphorus cycling and the effects of human activities
and climate change to simulate DOC flux transport in eastern North American rivers. To better quantify
riverine carbon transport processes at watershed scale, Yao et al. (2021) coupled the  scale-adaptive water
transport model (Li et al., 2013) to the DLEM model and applied the result to two mid-Atlantic
watersheds in the United States. Nevertheless, these models failed to consider the effects of
anthropogenic water regulation activities. Furthermore, constructing numerical simulation models is a



future development direction of riverine carbon flux estimation; at present, models are still not widely
used to simulate riverine carbon transport (Camino-Serrano et al., 2018).
In this study, we incorporated global soil and riverine DOC transport schemes considering
anthropogenic water regulation activities into Community Land Model 5.0 (CLM5.0) and conducted
numerical simulations at global scale (spatial resolution of about 1° for the land processes and 0.5° for
the river systems) during 1981–2013 to explore the impact of anthropogenic water regulation activities
on land-to-ocean riverine DOC transport.
**2. Model Development**
**2.1. Model Overview**
The model was developed based on CLM5.0, which is the land component of the CESM (Community
Earth System Model). CLM is widely used to simulate and study land surface ecohydrological processes,
surface energy exchange processes, and other biogeochemical processes. The latest version of CLM
updates most components of previous versions, explicitly represents land-use and land-cover change,
introduces a revised canopy interception parameterization, and uses the Model for Scale Adaptive River
Transport (MOSART, Li et al., 2013) to replace the original River Transport Model (RTM), in addition
to significant improvements in soil layer resolution, nitrogen cycle, and the snow model. Because the
scale of this study was global, the river transport model still uses linear RTM.
However, CLM5.0 lacks an expression of the soil DOC leaching process and the DOC transport and
transformation process in rivers. Therefore, in this paper, schemes for DOC leaching in soils and DOC
transport in rivers will be proposed and incorporated into CLM5.0 to simulate riverine carbon transport.
To investigate the effect of anthropogenic water regulation activities on global riverine DOC transport,
this study used the scheme proposed by Zeng et al. (2016), and coupled it with DOC transport processes.
The model framework is shown in Fig. 1.



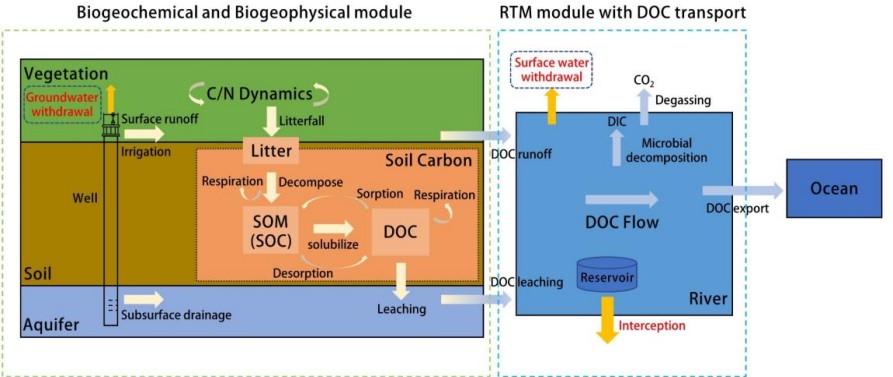

**Figure 1.** Schematic diagram of the land surface model with riverine dissolved organic carbon (DOC) transport and anthropogenic water regulation (C: carbon; N: nitrogen; SOM: soil organic matter; SOC: soil organic carbon; DIC: dissolved inorganic carbon).

**2.2. Soil DOC loss to the river**
Riverine DOC is mainly derived from organic carbon leaching processes in soil ecosystems in the
watershed. In CLM5.0, only the leaching process of soil mineral nitrogen is included, and therefore a
DOC production and loss process was introduced in this study. The soil biochemistry module in CLM5.0
was constructed based on the Century model (Parton et al., 1988), in which the decomposition of fresh
litter into soil organic matter is defined as a transformation cascade between the coarse woody debris
(CWD) pool, the litter pool, and the soil organic matter (SOM) pool. The NPP produced by plants
eventually enters the soil in the form of litter to constitute the soil carbon pool, accompanied by an
intervening loss through microbial heterotrophic respiration. Assuming that dissolved organic matter
(DOM) production is part of the turnover of litter pools and soil organic matter pools and is proportional
to soil water content, DOC production can be expressed as (Gerber et al., 2010):
$$P_{DOC,u \to d} = f_{DOM} \theta CF_{u \to d}, \qquad (1)$$

where $P_{DOC,u \to d}$ (g C m$^{-2}$ s$^{-1}$) is the DOC flux from the decomposition process; $f_{DOM}$ is the fraction
that enters the soil DOM pool; $\theta$ (m$^3$ m$^{-3}$) is the soil water content; and $CF_{u \to d}$ (g C m$^{-2}$ s$^{-1}$) is the
carbon flux from upstream to downstream carbon pools in the decomposition cascade.
Soil organic carbon remaining after plant growth and soil respiration is subject to loss as a dissolved
component leaching from the soil column. The leaching flux depends on the DOC concentration in the
soil water solution ($[DOC]$, g C kgH$_2$O$^{-1}$) and the hydrologic discharge rate from the soil column to
streamflow ($Q_{dis}$, kgH$_2$O m$^{-2}$ s$^{-1}$):





$$DOC_{leached} = [DOC]Q_{dis}k_{adsorb} - SR, \qquad (2)$$

where $[DOC]$ is calculated as:
$$[DOC] = \frac{NS_{DOC}}{WS_{tot\_soil}}, \qquad (3)$$

where $WS_{tot\_soil}$ (kgH$_2$O m$^{-2}$) is the total mass of soil water content integrated over the soil column and
$NS_{DOC}$ (g C m$^{-2}$) is the DOC in the soil pool.
Soil DOC readily complexes with metal ions in the soil and forms soil agglomerates, which enable
soil DOC to be adsorbed onto soil particles. The DOC adsorption coefficients can be estimated as (Li et
al., 2019; Neff & Asner, 2001):
$$k_{adsorb} = \frac{X_i}{X_i + RE}, \qquad (4)$$

$$RE = mX_i - b, \qquad (5)$$

where $X_i$ (mg g soil$^{-1}$) represents the initial DOC concentration and $m$ (dimensionless coefficient) and $b$
(mg g soil$^{-1}$) can be considered as measures of potential DOC sorption and desorption by soil.
The soil heterotrophic respiration flux of DOC, $SR$ (g C m$^{-2}$ s$^{-1}$), is estimated by an empirical function
(Janssens and Pilegaard, 2003):
$$SR = R_{10}Q_{s10}^{\frac{T-10}{10}}, \qquad (6)$$

where $T$ (℃) is the soil temperature; $R_{10}$ is the soil heterotrophic respiration flux at a soil temperature of
10°C; $Q_{s10}$ is the soil respiration temperature sensitivity.
It is necessary to limit the total DOC leaching flux at each time step so that it does not exceed the total
amount of DOC:
$$DOC_{leached} = \min\left(DOC_{leached}, \frac{NS_{DOC}}{\Delta t}\right). \qquad (7)$$

**2.3. Riverine DOC transport**
Soil DOC enters the river network system along with surface and subsurface runoff, where it is lost due
to processes such as microbial degradation. Therefore, based on the water transport framework, the large-
scale riverine DOC transport equation can be defined as:
$$\frac{dS_{DOC}}{dt} = F_{DOC}^{in} - F_{DOC}^{out} + R_{DOC} + L_{DOC} - k_{doc} * Q_{10}^{\frac{rt-20}{10}} * S_{DOC}, \qquad (8)$$

where $S_{DOC}$ (kg C) is DOC storage within the current grid cell; $R_{DOC}$ (kg C s$^{-1}$) and $L_{DOC}$ (kg C s$^{-1}$)
represent soil DOC runoff and leaching; $k_{doc}$ (s$^{-1}$) is the DOC decomposition rate in the river; $Q_{10}$



(=2.0) denotes the temperature coefficient; $rt$ (°C) represents the river water temperature, which is
calculated by a large-scale river water temperature model (Liu et al., 2020; van Vliet et al., 2012; Yearsley,
2009); $F_{DOC}^{in}$ (kg C s⁻¹) is the sum of inflows of riverine DOC from neighboring upstream grid cells;
and $F_{DOC}^{out}$ (kg C s⁻¹) is the riverine DOC flux leaving the current grid cell, which is calculated as follows:
$$F_{DOC}^{out} = \frac{vS_{DOC}}{d}, \tag{9}$$

where $v$ (m s⁻¹) is the effective riverine flow velocity, which is estimated by a simplified Manning's
equation (Oleson et al., 2013); $d$ is the Euclidean distance between two adjacent grid-cell centers.
**2.4. Anthropogenic water regulation**
Anthropogenic water regulation includes reservoir interception, surface water withdrawal, and
groundwater extraction and use. Because reservoir interception and surface water withdrawal are closely
related, they are together called surface water regulation. This study coupled the global reservoir
operation scheme (Hanasaki et al., 2006) with RTM using the method of Liu et al. (2020) to represent
the interception effect of reservoirs on runoff and solutes. The method assumed that the inflow from the
reservoir was the outflow from the current grid cell. Released flow from the reservoir was adjusted for
specific uses (flood control, irrigation, etc.), and surface withdrawals were deducted from the released
water.
Surface water is extracted directly from natural rivers and reservoirs to meet human water demands
(Liu et al., 2020; Wang et al., 2020; Xie et al., 2020):
$$S_{sw}{}' = S_{sw} - q_{sw}\Delta t, \tag{11}$$

where $S_{sw}{}'$ (mm) is the surface water storage after extraction; $S_{sw}$ (mm) is the original surface water
storage; $q_{sw}$ (mm s⁻¹) is the rate of surface water intake; $\Delta t$ denotes the model time step.
The groundwater extraction process can be expressed as (Zeng et al., 2016):
$$S_{gw}{}' = S_{gw} - q_{gw}\Delta t, \tag{12}$$

$$h' = h - \frac{q_{gw}\Delta t}{s}, \tag{13}$$

where $S_{gw}$ (mm) is the original unconfined aquifer water storage; $q_{gw}$ (mm s⁻¹) is the rate of
groundwater pumping; $h$ (mm) represents the original groundwater table depth; $s$ is the aquifer-
specific yield; $S_{gw}{}'$ (mm) and $h'$ (mm) denote the aquifer water storage and the groundwater table depth
after pumping.



Human water use can be divided into agricultural irrigation water and other industrial and domestic
water, where irrigation water is considered as effective precipitation directly back to the soil surface and
other water is directly added to the model surface runoff and evapotranspiration fluxes in a certain
proportion (Zou et al., 2015). This process can be estimated by the following equations:
$$q_{top} = q_{top} + q_{irrig},\tag{14}$$
$$q_{surf} = q_{surf} + 0.3q_{ind} + 0.3q_{dom},\tag{15}$$
$$q_{evap} = q_{evap} + 0.7q_{ind} + 0.7q_{dom},\tag{16}$$
where $q_{top}$ (mm s$^{-1}$) is the rate of net water flow entering the soil surface; $q_{surf}$ and $q_{evap}$ (mm s$^{-1}$) are
surface runoff and evaporation; and $q_{irrig}$, $q_{ind}$, and $q_{dom}$ (mm s$^{-1}$) denote irrigation, industrial, and
domestic water respectively.
**2.5. DOC transfer induced by water withdrawal and use**
Anthropogenic water regulation activities also affect DOC transport processes between land and river. It
was assumed here that (1) only the interception effect of reservoirs would be considered, ignoring the
migration transformation process in reservoirs, and the loss rate in reservoirs would be equal to that in
rivers; (2) because groundwater extraction usually occurs *in situ* and will pass through the filtering effect
of the soil layer, the part of DOC that returned to soil with groundwater extraction was ignored; (3) the
loss rate in the process of DOC returning to soil was equal to that in rivers.
The process of reservoir interception leading to retention of carbon in rivers can be expressed as:
$$F_{DOC,r} = \frac{v(con_r \Delta Q_r)}{d},\tag{17}$$
where $F_{DOC,r}$ (kg C s$^{-1}$) denotes the DOC flux retained by the reservoir; $con_r$ (kg C m$^{-3}$) is the DOC
concentration in the reservoir; $\Delta Q_r$ (m$^3$) is the water volume change in the reservoir.
The DOC flux extracted from surface water is calculated based on the intake rate and the solute
concentration in the current grid cell and enters the soil DOC pool after irrigation. The reduction in soil
DOC leaching due to groundwater extraction is then calculated based on soil DOC concentration and
groundwater pumping rate.



## 3. Data and Experimental Design

### 3.1. Data Sources

The climate input forcing data set ($0.5° \times 0.5°$) used for the model proposed in this study was obtained from CRU-NCEP Version 7 (Viovy, 2018), including air temperature, humidity, incoming solar radiation, precipitation, surface pressures, and surface winds. The basic land-surface datasets required to drive the model were set up using the default CLM 5.0 settings with a spatial resolution of $0.9° \times 1.25°$; more details are available in the technical notes (Lawrence et al., 2018). The global monthly mean atmospheric $CO_2$ concentration dataset came from the NOAA/Earth System Research Laboratory (https://www.esrl.noaa.gov/gmd/ccgg/trends/global.html).

Reservoir information was obtained from the Global Reservoir and Dam Database (GRanD, Lehner et al., 2011), containing information on 6,862 dams and their associated reservoirs worldwide, and interpolated to a spatial resolution of $0.5° \times 0.5°$.

The human water use activity dataset was derived from the global long-term surface and groundwater withdrawal dataset estimated by Liu et al. (2020). The dataset has a spatial resolution of $0.5° \times 0.5°$ and contains agricultural, industrial, and domestic water demands from 1958 to 2017.

### 3.2. Observation Data

Because there are few datasets of long time-series observations of DOC fluxes for large global rivers, annual averages were used to validate the model simulations. The dataset was derived from the database developed by Dai et al. (2012), which provides discharge and DOC flux observations for sites on the world's major large rivers. These sites were globally distributed and were influenced by various climatic and human activities.

### 3.3. Experimental Design

To investigate the effect of anthropogenic water regulation on DOC transport in rivers, three sets of simulations were designed using the developed model (Table 1). The first simulation (CTL) was a control experiment without considering any anthropogenic water regulation activities. The second simulation (EXPA) only considered surface water regulation, and the last simulation (EXPB) considered all anthropogenic water regulation. All simulations were run from 1981 to 2013 with a spatial resolution of $0.9° \times 1.25°$ for the land-surface module and $0.5° \times 0.5°$ for the RTM. The results were output on a





monthly scale. Before the formal numerical simulations, the 1901–1920 atmospheric forcing data cycle
was used to drive the model without any anthropogenic water regulation as the spin-up run to reach an
equilibrium state.

**Table 1.** Experimental design

| Name | Period | Surface regulation | Groundwater regulation |
|------|--------|--------------------|------------------------|
| CTL  | 1981–2013 | ✗ | ✗ |
| EXPA | 1981–2013 | ✓ | ✗ |
| EXPB | 1981–2013 | ✓ | ✓ |

**4. Results**
**4.1. Model Evaluation**
Figure 2 shows the spatial distribution of multi-year average soil DOC losses, which are the sum of DOC
surface runoff and subsurface leaching. The results show that the global distribution of soil DOC losses
varied widely, especially in Russia and Southeast Asia, western Africa, and tropical South America,
where the losses exceeded 18,000 kg C km$^{-2}$ yr$^{-1}$, whereas low runoff arid regions such as northwestern
China, India, and North Africa had the smallest soil DOC losses. The tropics and the temperate regions
of the Northern Hemisphere were the regions with the highest soil DOC losses, which was generally
consistent with previous studies (Harrison et al., 2005).

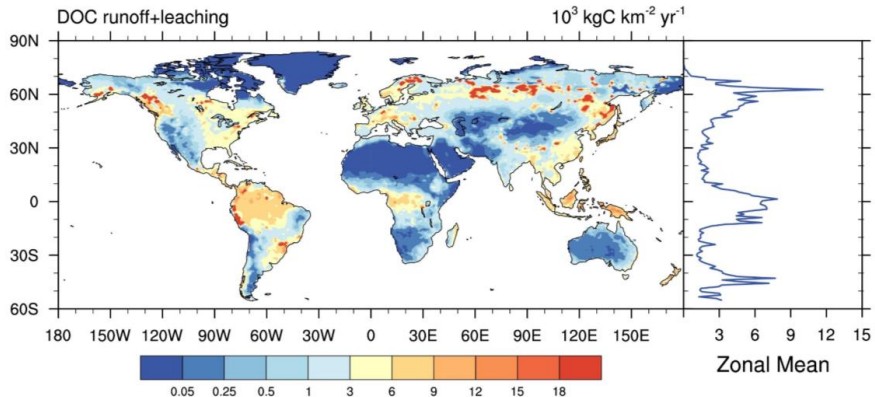

**Figure 2.** Spatial distribution and zonal mean of multi-year average soil DOC losses from 1981 to 2013.

The multi-year average river discharges and DOC export fluxes simulated by the developed model



were then compared with observed data. Because the model resolution was 0.5° × 0.5°, only 106 rivers
with watershed areas larger than 2,500 km² were selected. The simulated river discharges were slightly
overestimated (Fig. 3c), but fit well with observations (Fig. 3a) and provided a solid basis for subsequent
simulation of river carbon exports. In addition, the simulated riverine DOC export fluxes tended to be
overestimated in temperate regions and underestimated in the tropics (Fig. 3d), but were close to the 1:1
line compared to the observed DOC fluxes, with $R^2$ reaching 0.61 and significantly correlated (Fig. 3b).
Moreover, the total global river DOC export fluxes simulated by the proposed model were compared
with the results of previous studies. We estimated that the global terrestrial ecosystem delivers about
199.78 Tg of DOC per year to the ocean via rivers, which was in the middle of the values derived from
previous studies (Table 2). Therefore, it could be believed that the model has reasonable accuracy and
can be applied to global-scale riverine DOC export simulation studies.

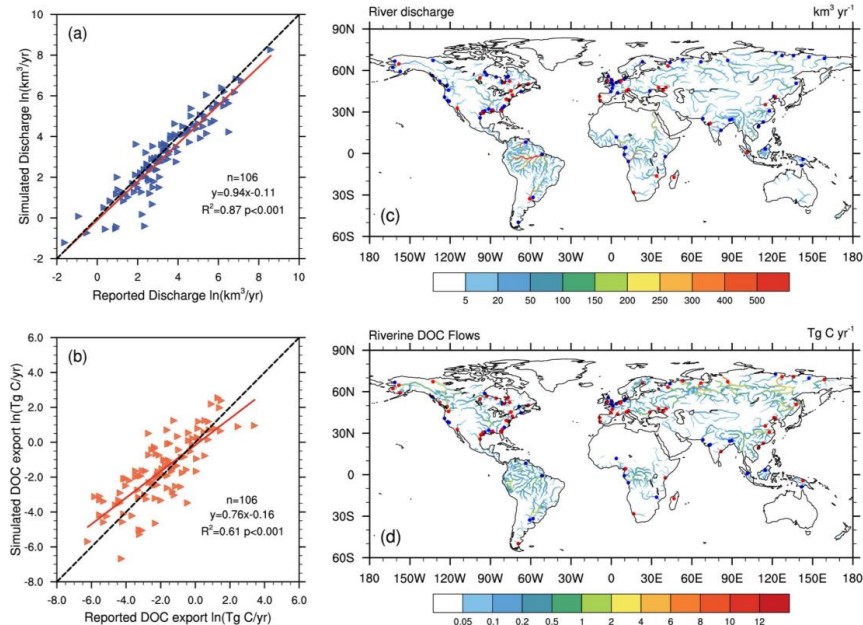

**Figure 3.** Simulated and reported annual (a) river discharge and (b) riverine DOC export flux for 106 global rivers. Spatial distributions of (c) annual discharge and (d) annual riverine DOC exports during 1981–2013. The dots in the map correspond to the locations of the 106 river sites, where blue dots indicate sites that are simulated underestimates and red dots indicate sites that are simulated overestimates.



**Table 2.** Comparison of simulated global total riverine DOC export fluxes with previous studies

| Method | DOC (Tg C yr$^{-1}$) | Data Source |
| --- | --- | --- |
| GEMS-GLORI | 215 | Meybeck (1982) |
| Empirical model | 204 | Smith & Hollibaugh (1993) |
| Empirical model | 204.81 | Ludwig et al. (1996) |
| Global C: N | 361 | Aitkenhead & McDowell (2000) |
| NEWS-DOC | 170 | Harrison et al. (2005) |
| Global-NEWS | 170 | Seitzinger et al. (2005) |
| Statistical estimation | 246 | Cai (2011) |
| TRIPLEX-HYDRA | 240 | Li et al. (2019) |
| Empirical model | 131.6 | Fabre et al. (2020) |
| CLM5.0-RTM | 199.78 | This study |

**4.2. Effects of surface water regulation on riverine DOC transport**
The difference between EXPA and CTL was used to obtain the effect of surface water regulation on land
surface hydrological variables. Surface water use has resulted in changes in latent and sensible heat fluxes
in most global irrigation water-using regions (Fig. 4a, 4b), especially in arid or semi-arid regions such as
northern China, India, and the central United States, where latent heat fluxes have increased and sensible
heat fluxes have decreased. Soil and surface temperatures in these regions have also decreased due to the
cooling effect of irrigation (Fig. 4c, 4d). Figure 4e shows that irrigation led to an overall increase in soil
moisture, especially in northern India, Western Europe, and the midwestern United States. In addition,
irrigation also led to an increase in total runoff (Fig. 4f).





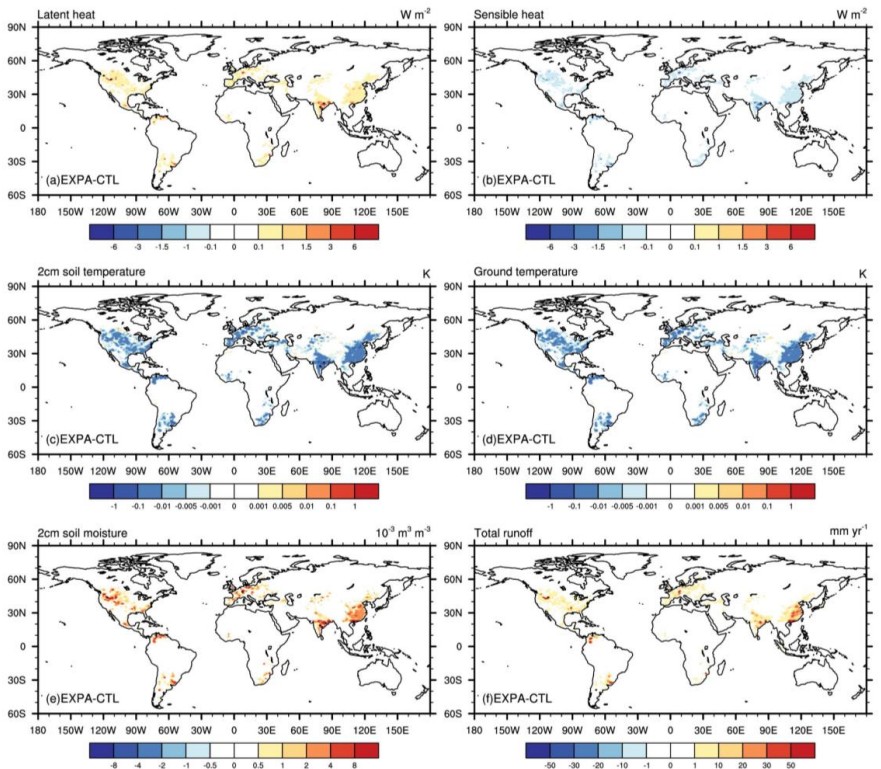

**Figure 4.** Spatial distribution of multi-year average changes in land surface hydrological variables due to surface water regulation from 1981 to 2013: (a) latent heat flux, (b) sensible heat flux, (c) 2 cm soil temperature, (d) surface temperature, (e) 2 cm soil moisture, (f) total runoff. The black dots are the regions that pass the significance *t*-test at the 95 % confidence level.

Figures 5a and 5b display the effects of surface water regulation on soil carbon losses. Specifically,
the hotspots of significantly increased surface DOC runoff were in areas of high agricultural influence,
such as the central United States, northern India, and northern and eastern China, reaching up to 2,000
kg C km$^{-2}$ yr$^{-1}$, but the increase in subsurface leaching was relatively small. This may have been the case
because surface water withdrawals from rivers and reservoirs were returned to the soil by irrigation,
bringing back some DOC, directly increasing surface runoff, and also increasing subsurface runoff, and
thus increasing soil DOC losses.

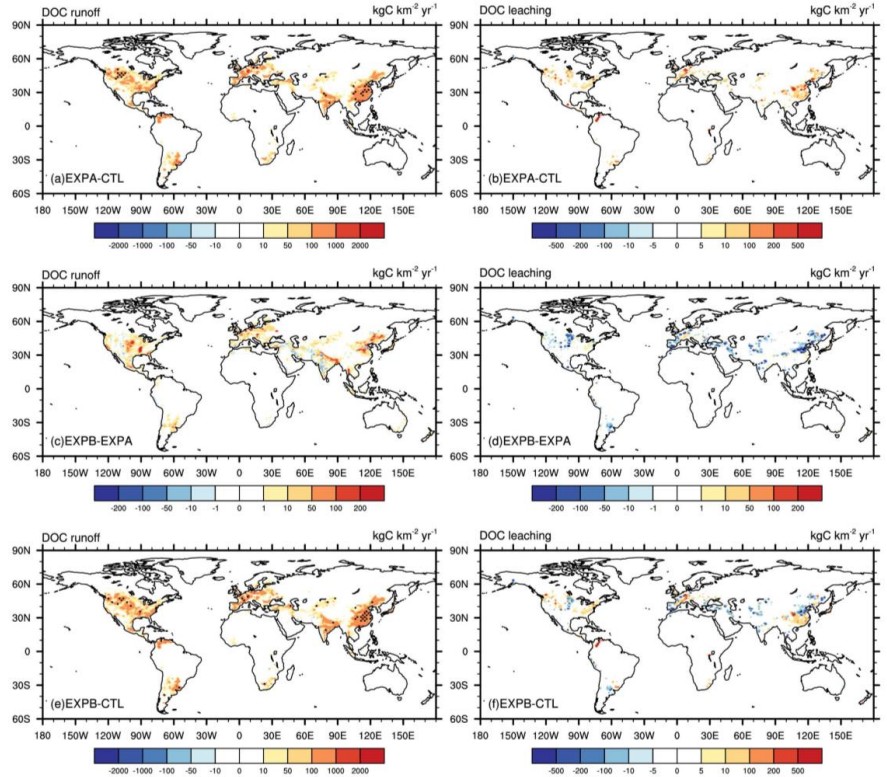

**Figure 5.** Spatial distribution of multi-year average changes in soil carbon losses due to surface water regulation (a, b), groundwater regulation (c, d), and anthropogenic water regulation (e, f) from 1981 to 2013. The black dots are the regions that pass the significance *t*-test at the 95 % confidence level.

From Fig. 6a and Fig. 6b, surface water regulation had a significant effect on river discharge and
riverine DOC flow. The combined effects of reservoir interception and surface water withdrawal reduced
the discharge and DOC export of most rivers globally, with significant reductions of more than 50 Gg C
yr$^{-1}$ in the Yangtze, Yellow, Mississippi, and Ganges Rivers and in some basins in Western Europe. Some
rivers in northern South America experienced increased riverine DOC export, but not significantly,
probably because the increase in river flow caused by agricultural irrigation could have been greater than
the decrease caused by surface water regulation.



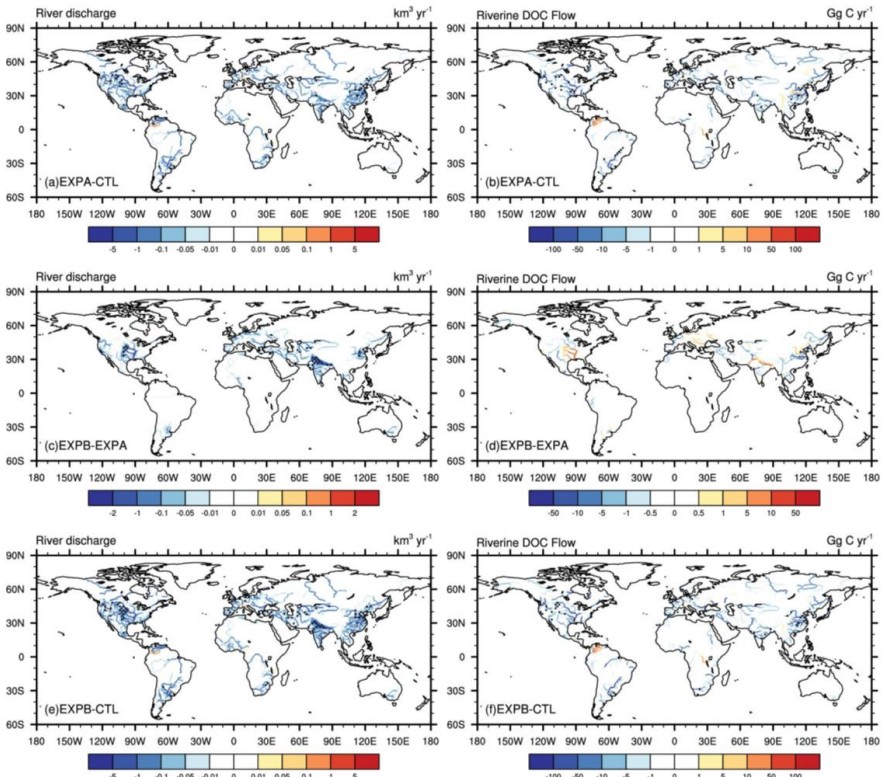

**Figure 6.** Spatial distribution of multi-year average changes in river discharge and riverine DOC flow due to surface water regulation (a, b), groundwater regulation (c, d), and anthropogenic water regulation (e, f) from 1981 to 2013. The black dots are the regions that pass the significance *t*-test at the 95 % confidence level.

The blue line in Fig. 7 represents the time-series variation of surface water regulation on global riverine
organic carbon to the ocean. Surface water regulation greatly reduced global riverine DOC transport to
the ocean, from –11.1 Tg yr$^{-1}$ in 1981 to –16.4 Tg yr$^{-1}$ in 2013 (Fig. 7a), with a multi-year average
retention efficiency of about 6 %. This may be related to the fact that reservoir interception increases the
residence time of water and thus increases DOC removal rate (Liu et al., 2022). The regions most affected
by surface water regulation were the Pacific and Atlantic Oceans, and as surface water use in these
regions became more frequent, the reduction in DOC delivery to the ocean was intensified each year.
There was no significant change in the Arctic Ocean region, which may have been due to less
anthropogenic disturbance in the alpine region.

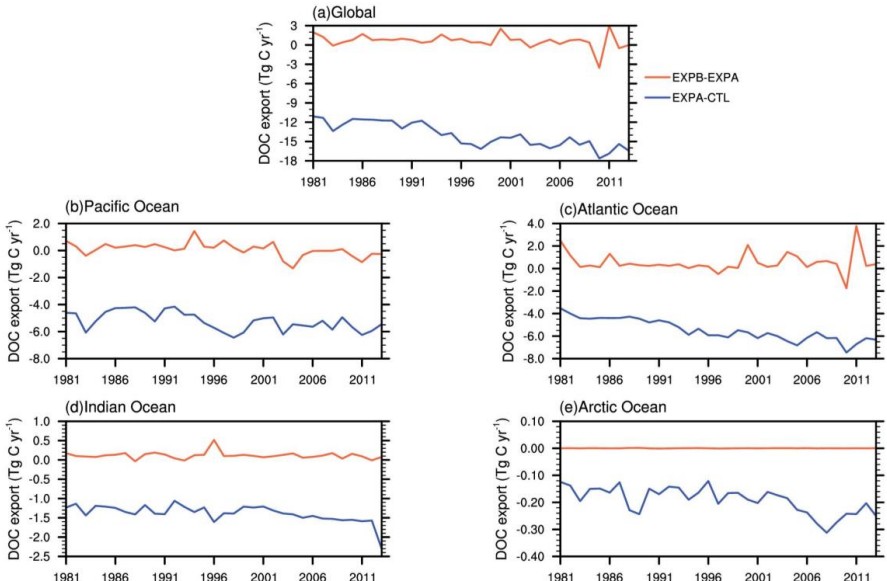

**Figure 7.** Time series of changes in DOC export to oceans due to surface water (blue line) and groundwater regulation (orange line) from 1981 to 2013: (a) global, (b) Pacific Ocean, (c) Atlantic Ocean, (d) Indian Ocean, (e) Arctic Ocean.

### 4.3. Effects of groundwater regulation on riverine DOC transport

The effects of groundwater regulation on land surface hydrological variables were obtained using the difference between EXPB and EXPA, as shown in Fig. 8. It can be seen that groundwater extraction increased latent heat fluxes, decreased sensible heat fluxes, decreased soil and surface temperatures, and increased soil moisture in most regions of the world. The most significant impacts were in northern China, northern India, Pakistan, and the central United States, where climate conditions are dry and groundwater extraction is frequent. Unlike surface water regulation, groundwater extraction has a negative impact on total runoff (Fig. 8f). Because groundwater is extracted from underground aquifers, whereas surface water is extracted from rivers and reservoirs, surface water use directly increases total land surface runoff. However, the impact of groundwater extraction on runoff depends on the groundwater pumping rate, infiltration rate, and soil evaporation capacity. The increase in latent heat flux leads to an increase in surface evapotranspiration, which results in a decrease in runoff.

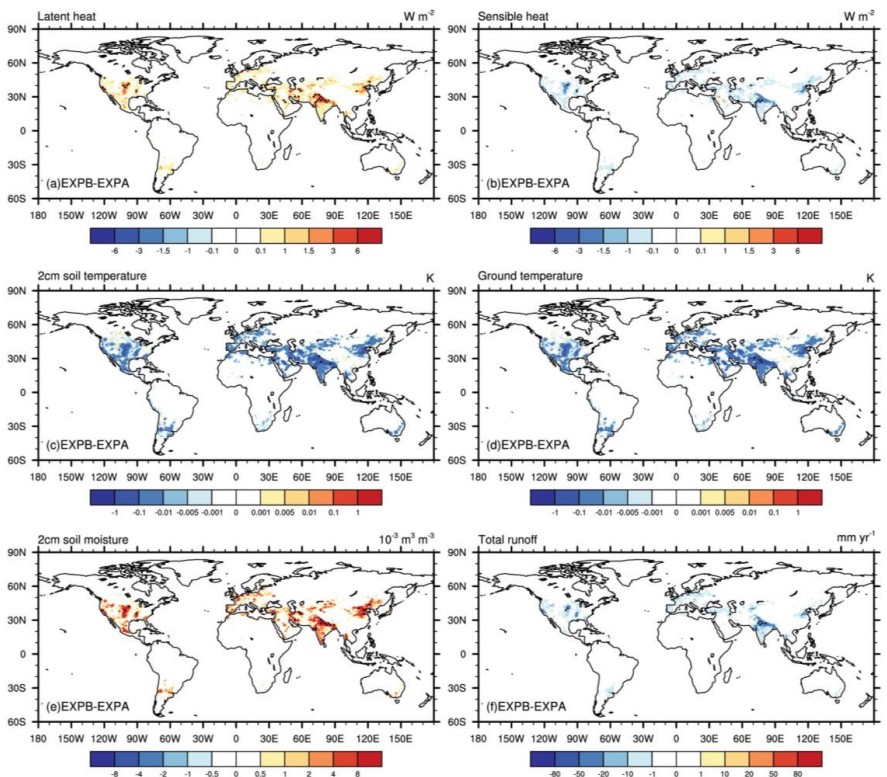

**Figure 8.** Spatial distribution of multi-year average changes in land surface hydrological variables due to groundwater regulation from 1981 to 2013: (a) latent heat flux, (b) sensible heat flux, (c) 2 cm soil temperature, (d) surface temperature, (e) 2 cm soil moisture, (f) total runoff. The black dots are the regions that pass the significance *t*-test at the 95 % confidence level.

Figures 5c and 5d show the effect of groundwater regulation on soil carbon losses. On the one hand,
extracting water from underground aquifers led to a reduction in subsurface runoff and a consequent
reduction in DOC leaching, especially in northern China and the central United States, where DOC
leaching reductions reached 200 kg C yr$^{-1}$. On the other hand, groundwater irrigation led to an increase
in surface runoff, which led to an increase in DOC runoff. The most affected areas are characterized by
well-developed agriculture.
Figures 6c and 6d show the spatial distribution of the effects of groundwater regulation on river
discharge and DOC export from 1981 to 2013. It can be seen that river discharge significantly decreased
in areas with high groundwater extraction rates, such as the central United States, Pakistan, Afghanistan,
and northern China, resulting in a decrease in riverine DOC export. The largest decrease occurred in the



Yangtze River Basin in China, reaching 50 Gg C yr$^{-1}$; most other rivers were around 10 Gg C yr$^{-1}$. In addition, although river discharge was reduced in some river sections, soil DOC loss was higher, and DOC export fluxes were still increasing, especially in the lower Yellow River, Mississippi River, and Ganges River basins. This was due to the predominance of agricultural irrigation water in these regions.

The amount of carbon flux variation influenced by groundwater regulation was relatively small compared to that influenced by surface water regulation, but there was some interannual fluctuation, with the greatest impact during 2009–2012 (Fig. 7). The intermittent increase and decrease of the variation indicate that river carbon transport fluxes did not decrease directly with increases of groundwater pumping rate, but were also related to the complex carbon and nitrogen cycling processes in terrestrial ecosystems. In addition, irrigation after groundwater extraction from an underground aquifer did not consider directly sending DOC back to the soil carbon pool, and therefore the carbon flux changes were smaller. Because groundwater regulation activities are mostly concentrated in the northern temperate zone, the Pacific and Atlantic regions were the most obviously affected, whereas the remaining regions did not change much.

**4.4. Effects of anthropogenic water regulation on riverine DOC transport**

This section discusses the combined effects of anthropogenic water regulation on soil and riverine carbon transport using the EXPB minus CTL results. The effects of anthropogenic water regulation on total runoff both increased and decreased globally (Fig. 9f). The western United States, Venezuela, and northern China showed an increase in runoff due to the high intensity of irrigation water use in agriculture. In contrast, regions such as northern India and the central United States showed a decrease in runoff due to frequent groundwater extraction. Overall, human water regulation activities led to an increase in latent heat fluxes and soil moisture and a decrease in sensible heat fluxes and in soil and ground temperatures.

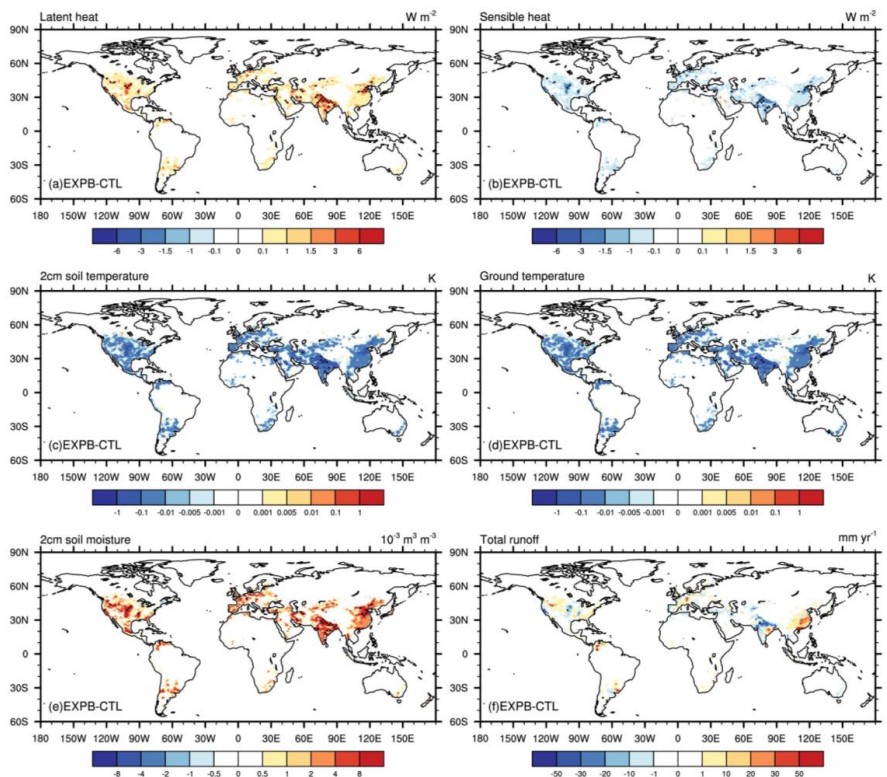

**Figure 9.** Spatial distribution of multi-year average changes in land surface hydrological variables due to anthropogenic water regulation from 1981 to 2013: (a) latent heat flux, (b) sensible heat flux, (c) 2 cm soil temperature, (d) surface temperature, (e) 2 cm soil moisture, (f) total runoff. The black dots are the regions that pass the significance *t*-test at the 95 % confidence level.

Figure 5e shows that soil DOC runoff increased, especially in northern China and the midwestern
United States. DOC leaching decreased in some river sections (Fig. 5f), but not significantly. Although
soil DOC runoff showed an overall increase, DOC export fluxes decreased in most rivers globally due to
water regulation (Fig. 6f). On the one hand, human water use activities led to a decrease in river discharge
(Fig. 6e), and on the other hand, reservoir have intercepted part of riverine DOC, which led to an increase
in microbial activity, resulting in a decrease in river carbon flux. In contrast, in the Mississippi and
Ganges River basins, although groundwater regulation increased their DOC export fluxes (Fig. 6d), they
still showed a decrease under the negative feedback effect of surface water regulation, indicating that
most rivers globally are mainly influenced by reservoir interception and surface water withdrawal.
Five typical rivers were selected to exhibit how anthropogenic water regulation affects monthly and



annual average DOC flows in rivers. The selected rivers were the Mississippi River in the United States,
the Danube River in Europe, the Ob River in Russia, the Yangtze River in China, and the Ganges River
in India. Figure 10 displays the seasonal and interannual variation of DOC flow rates in the five rivers as
calculated by the three sets of simulations respectively. Anthropogenic water regulation had a significant
impact on the Mississippi, Danube, Yangtze, and Ganges Rivers, which decreased significantly in winter
and early spring, whereas the Ob River was almost unaffected. This was the case because of weak water
management activities in the Ob River, whereas the other subtropical and temperate rivers had intense
water management activities and significant seasonal variation in runoff. In addition, only the Mississippi,
Yangtze, and Ganges rivers were affected by minor groundwater regulation, usually occurring during dry
periods, whereas in most seasons, the rivers were affected only by surface water regulation (including
reservoir interception). The annual results showed a significantly strengthening trend of riverine DOC
reduction due to the influence of anthropogenic water regulation, especially in the Danube and Yangtze
Rivers, where the retention percentage in 2013 was four to five times higher than in 1981, up to more
than 50 %, indicating a clear intensification of human water management activities. The influence on the
Mississippi and Ganges Rivers increased slightly and stabilized at about 30–40 %, whereas the influence
on the Ob River was almost 0.

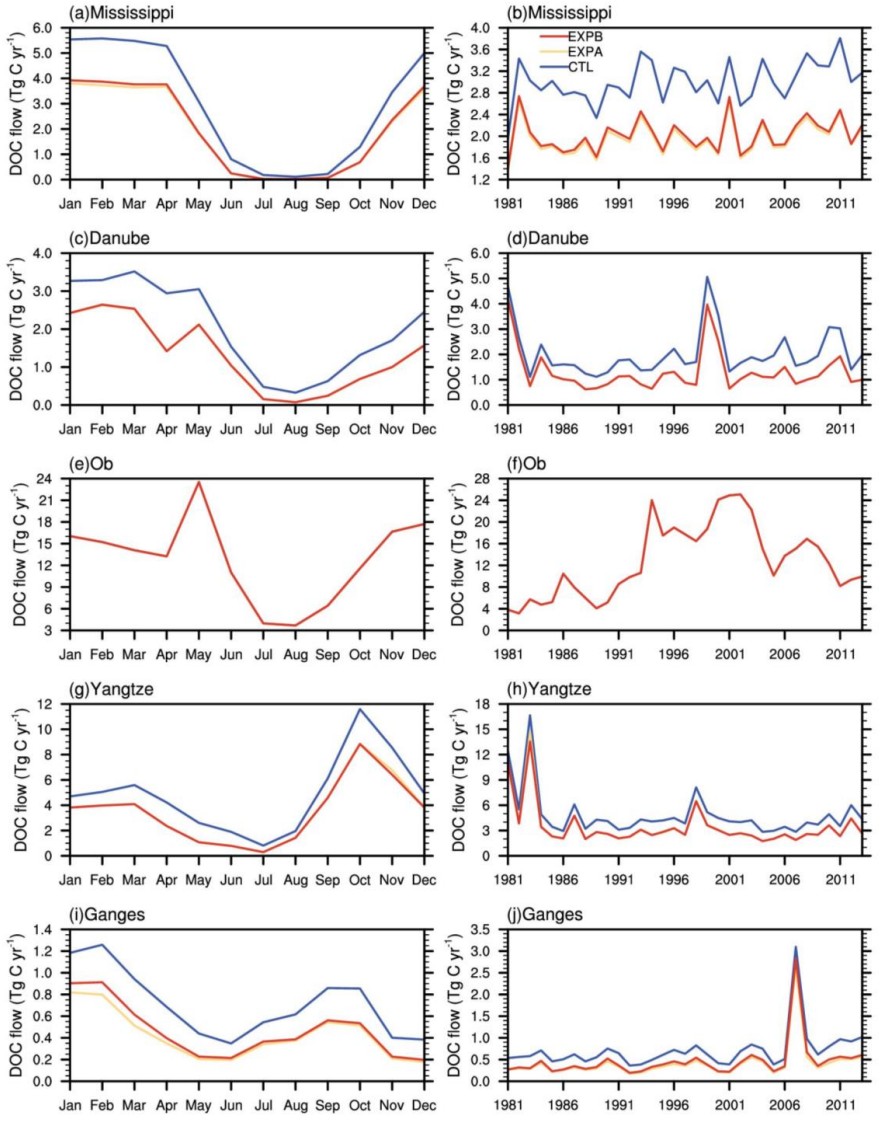

**Figure 10.** Time series of monthly and annual average riverine DOC flow rates for the five typical rivers simulated by CTL (blue line), EXPA (yellow line), and EXPB (red line): (a, b) Mississippi River (32.25° N, 91.25° W), (c, d) Danube River (45.25° N, 28.75° E), (e, f) Ob River (66.25° N, 66.75° E), (g, h) Yangtze River (30.75° N, 117.75° E), (i, j) Ganges River (24.25° N, 88.25° E).

Riverine DOC export fluxes have obvious spatial heterogeneity. Six zones were defined according to
the latitudes where the river mouths are located, and the effects of the presence or absence of
anthropogenic water regulation on DOC export fluxes are shown in Fig. 11. The hotspot regions of



riverine DOC export are concentrated in the tropics (23.5° S–23.5° N) and the mid and high latitudes of
the Northern Hemisphere (40–90° N). The DOC export fluxes of rivers between 40° N and 66° N
accounted for 35.32 % of total global export flux. Due to anthropogenic water regulation, the global DOC
export flux was reduced by 13.36 Tg C yr$^{-1}$ compared to the case with no human regulation, with the
greatest impact concentrated in the subtropical and temperate regions of the Northern Hemisphere (23.5–
66° N) because this is the region with the highest intensity of human water use activity.

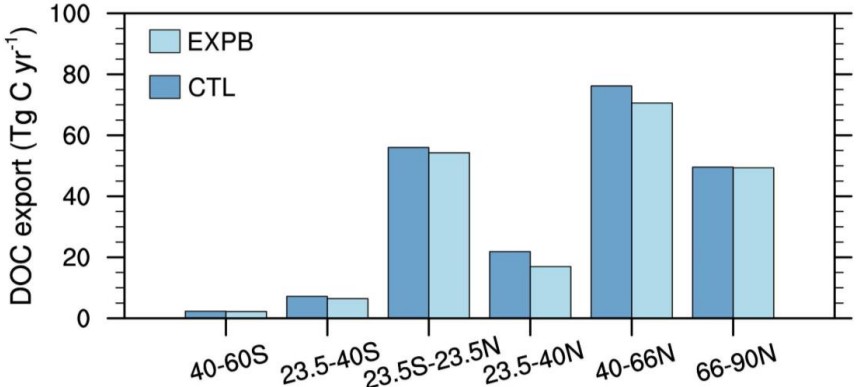

**Figure 11.** Bar chart of latitudinal band distribution of multi-year average DOC export fluxes from 1981 to 2013. Dark blue indicates no human activity, and light blue indicates anthropogenic water regulation.

Overall, anthropogenic water regulation reduced global riverine carbon fluxes, and the reduction in
DOC fluxes also intensified over time, from –9.13 Tg C yr$^{-1}$ to –16.45 Tg C yr$^{-1}$ (Fig. 12), and the
reduction percentage also increased from 4.83 % to 6.20 %. Rivers in the Pacific and Atlantic regions
were more affected by water regulation, and the interannual changes were more consistent with the global
picture. The flux of rivers into the Indian Ocean, which was reduced by water regulation, was about 1.27
Tg C yr$^{-1}$, which was small compared to the global flux, and the flux into the Arctic Ocean was almost
negligible due to the scarcity of human activities.

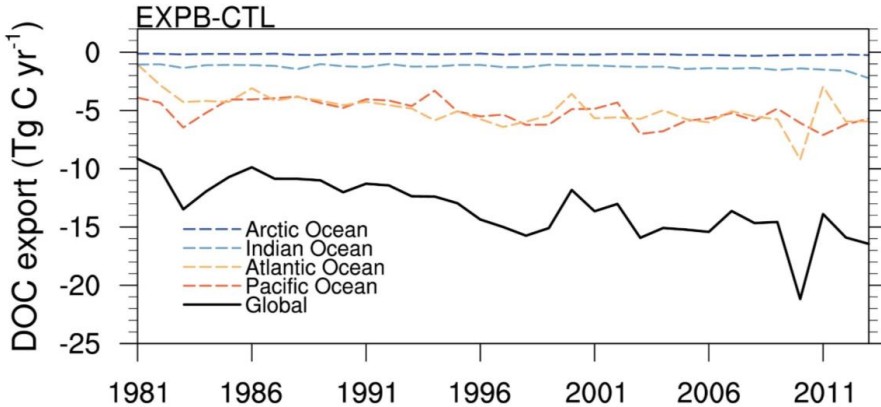

**Figure 12.** Interannual variability in the impact of anthropogenic water regulation on riverine DOC delivery from rivers to the ocean.

**5. Conclusions**
This study has developed schemes that consider soil and riverine DOC dynamics and anthropogenic
water regulation activities and has incorporated them into the land surface model CLM5.0. The simulated
river discharges and riverine DOC export fluxes were in good agreement with observations obtained for
106 major world rivers. Surface water and groundwater use datasets were used as inputs to the model,
and three sets of numerical simulations were conducted from 1981 to 2013 on a global scale to investigate
the effects of anthropogenic water regulation on riverine DOC transport.
The main conclusions of this study are as follows. First, anthropogenic water regulation activities
increased soil losses in most arid and semi-arid regions of the world, although groundwater extraction
reduced subsurface runoff and decreased DOC leaching; however, this decrease was less than the increase
in DOC runoff due to irrigation. Second, the DOC export fluxes of the Yangtze, Yellow, Mississippi, and
Ganges River basins were significantly reduced by reservoir regulation and surface water withdrawal.
However, DOC export fluxes in these areas showed an increase under groundwater regulation, but the
increase was small, indicating that DOC transport in most rivers globally is mainly influenced by
reservoir interception and surface water regulation. Third, further analysis showed that subtropical and
temperate rivers with intensive water management regimes were more affected and that DOC flows
decreased substantially in winter and early spring. The retention percentage has been increasing year by
year, up to over 50 %, indicating a clear intensification of human water management activities, especially



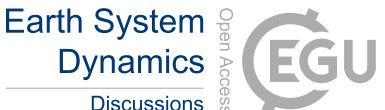

along the Danube and Yangtze Rivers. In addition, the greatest impact of anthropogenic water regulation
activities was concentrated in the region from 23.5°N to 66°N because this zone contains the highest
intensity of human water use activities. Fourth, global riverine DOC flux transport to the ocean decreased
by an average of 13.36 Tg C yr$^{-1}$ per year due to anthropogenic water regulation activities, and the
decrease in DOC flux became more pronounced with time, from -9.13 Tg C yr$^{-1}$ (4.83 %) in 1981 to –
16.45 Tg C yr$^{-1}$ (6.20 %) in 2013, especially in the Pacific and Atlantic Ocean regions. Meanwhile, the
Arctic Ocean region was almost unaffected due to low anthropogenic disturbance.

In general, this study has developed an effective scheme to simulate DOC export from terrestrial to

aquatic systems, which is important for improving carbon budget estimation and integrated ecosystem
management.

***Code and Data Availability.*** The observed river discharge and riverine DOC exports data can be available
through Dai et al. (2012). The source code of CLM 5.0 is available online
([https://www.cesm.ucar.edu/models/clm](https://www.cesm.ucar.edu/models/clm)). The FORTRAN code of developed model in this study is
available upon request. Please contact Zhenghui Xie at [zxie@lasg.iap.ac.cn](mailto:zxie@lasg.iap.ac.cn). The drawing language is the
NCL language.

***Author contributions.*** The scientific framing of this paper was developed by YY, ZX, BJ. The model
was initiated by YY and YW. The literature review was performed by HY, YT and SC. Analyses and
scientific post-processing were performed by LW and RL. All authors discussed the results and
contributed to the writing of the paper.

***Competing interests.*** The contact author has declared that neither they nor their co-authors have any
competing interests.

***Acknowledgements.*** This work was jointly supported by the National Natural Science Foundation of
China (grant number: 41830967), the National Key Research and Development Program of China (grant
number: 2022YFC3201903), and the Youth Innovation Promotion Association CAS (2021073).



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
