# Peer review of "Impacts of anthropogenic water regulation on global"

_Earth System Dynamics, 2023_

## Author Comment (AC1)

**Response to Anonymous Referee #1:**

Dear Referee #1,

Thank you very much for your time involved in reviewing the manuscript and your constructive suggestions. To facilitate this discussion, we first retype your comments in **bold font** and then present our responses to the comments. An annotated version of the revised manuscripts is attached.

**This manuscript incorporated a riverine dissolved organic carbon transport scheme to the land surface model CLM5.0 to evaluate the impacts of anthropogenic water regulation on riverine DOC discharges and transport. The paper is well written, presenting an interesting work in a clear and organized way. I have a few minor comments below.**

*Response:* We appreciate your very encouraging comments on the merits, and hope that the response has fully addressed all your concerns.

1. **Equation (2): Please denote the unit of DOC leaching flux.**

*Response:* We will add the unit of DOC leaching flux ($g\ C\ m^{-2}\ s^{-1}$).

2. **Line 176: "Riverine DOC is mainly derived from organic carbon leaching processes in soil"; some literature support is required here.**

*Response:* We will add corresponding references as suggested.

3. **Line 189-190: where is the reference for choosing this weighting coefficient?**

*Response:* We chose this coefficient according to the previous studies (Liu et al., 2019; Zou et al., 2014), and we will add corresponding references at the same time.

4. **Section 3.1: I suggest adding a table to show the main datasets used for model running and validation in this study.**

*Response:* Thanks for your valuable suggestion. We will add a table to summarize the main datasets used in this study.

5. **Line 221: Please introduce the details for the human water use activity dataset. A description of what data sources were used?**

*Response:* Based on the comment, we will revise the manuscript. The human water use activity dataset was derived based on five datasets: the water use dataset from the Food and Agricultural Organization (FAO), a shape file data of national boundaries, the Global Map of Irrigation Areas, version 5 (GAMIP5; Siebert et al., 2013), the historical monthly soil moisture levels and saturated soil moisture levels(Zeng et al., 2017), and the FAO water information system for 2010, which contained the agricultural, industrial, and municipal water withdrawals.

[Figure]

6. **Line 205: Only the fluxes into the soil carbon pool after surface water extraction are described. What about groundwater extraction?**

*Response:* Because groundwater extraction usually occurs in situ and will pass through the filtering effect of the soil layer, we hypothesized the part of DOC that returned to soil with groundwater extraction was ignored in our parameterization scheme.

7. **In Section 2.1, the parameters mentioned in the developed soil and river carbon dynamics parameterization scheme are uniform or spatially varying?**

*Response:* The parameters mentioned in our developed schemes are uniform. In fact, it does not correspond to spatial heterogeneity, and we will further refine and modify the parameterization scheme in our future work. The current parameterization scheme has reasonable accuracy in the simulation results, so we believe that our model can be applied to global-scale riverine DOC transport simulation studies.

8. **Line 252: Figures 3a and 3c seem to underestimate. Please check carefully and modify.**

*Response:* Thank you so much for your careful check. We will modify it in our manuscript.

9. **Line 273: Are constants (0.3 and 0.7) in equations the same for the whole world?**

*Response:* Yes, we set the constant due to the limitation of data.

10. **This study developed a model to describe the soil carbon leaching and riverine carbon transport processes, which are not well described in previous land surface models. But the discussion of current uncertainties and limitations in modeling is missing. It should be discussed more.**

*Response:* Thank you for your advice. We will add some discussion of current uncertainties and limitations in our manuscript.

11. **Line 354-355: The authors state that the three rivers were affected by minor groundwater regulation. Please briefly explain the impact and the reasons.**

*Response:* In our selected rivers, only the Mississippi, Yangtze, and Ganges rivers were affected by minor groundwater regulation, which usually occurred during the dry period, where DOC export increased slightly in the Mississippi and Ganges rivers because of higher soil leaching due to irrigation, while DOC export decreased in the Yangtze River due to a significant reduction in river discharge. This also corresponds to the results in Section 4.3.

12. **In section 5, some words about future work are needed.**

*Response:* Thank you for your suggestion. We will add some discussion about future work in our manuscript.

Once again, thank you very much for your comments and suggestions.

Sincerely,

Yanbin You, Zhenghui Xie, Binghao Jia, et al.

---

## Author Comment (AC2)

**Response to Anonymous Referee #2:**

Dear Referee #2,

Thank you very much for your helpful comments and suggestions to improve our manuscript. To facilitate this discussion, we first retype your comments in **bold font** and then present our responses to the comments.

**This study aims to assess the effects of anthropogenic regulation of waters on the global transport of DOC by rivers. Given the importance of DOC in the carbon cycle and the potential of human activities to alter its cycling, this is a very important aim. I find the paper generally interesting, but have a few concerns:**

*Response:* We appreciate your clear and detailed feedback, and hope that the response has fully addressed all your concerns.

**The presentation can be made clearer. The authors should make an effort to make it more accessible to non-modelers (like me), and to readers who want to take home the message without detailed reading of the methods. For example, the different simulations of control conditions and different parts of water regulation considered (CTL, EXPA, EXPB) are in several of the figures presented without explanation or spelling out.**

*Response:* Thank you very much for your suggestion. We will modify the subtitle and legend of Fig. 4~12 to make it easier for the reader to understand.

**Is there some way to add (or more carefully discuss) uncertainty ranges around the various estimates and graphs? The current version presents and compares several numbers with 3-4 significant digits, with no confidence intervals.**

*Response:* Based on your comment, we will add a standard deviation after the estimated value to indicate its uncertainty range (mean±std).

**Table 2 could be expanded, it appears incomplete. There are several additional estimates of DOC export (possible resulting in a higher median than presented in the manuscript). Some (but not all) are cited in Drake et al. 2018 (Limnol Oceanogr Letters).**

*Response:* Thanks for your detailed comment. We have read the article you mentioned and found that most of the carbon flux which export to the ocean is estimated to be 0.95 Pg C yr$^{-1}$, but it includes all forms of carbon in rivers. We set the organic carbon (OC) / inorganic carbon (IC) ratio to 0.4/0.5, in which 55% of OC flux is dissolved (DOC). Finally, we calculated that the riverine DOC export flux was 232.22 Tg C yr$^{-1}$. We will add this result to the table at the same time. Besides, we will also add another result from van Hoek et al. (2021, Environ. Sci. Technol.).

**I do not understand how transformations in the regulated and unregulated waters are treated. The methods (line 196) say that "migration transformation" is ignored in the model, and loss rate is assumed equal in reservoirs and rivers. In contrast, one of the model results (line 287) is suggested to be due to increased residence time by the construction of reservoirs, causing increased DOC removal. How is this compatible?**

*Response:* Thank you so much for your careful check. We will revise this sentence to "This may be related to the fact that the reservoir adjusting the river discharge and intercepting the riverine DOC."

**Line 292: "alpine" should be "arctic"**

*Response:* We will revise it in our manuscript.

Thank you again for your detailed and precious comments on our manuscript.

Sincerely,

Yanbin You, Zhenghui Xie, Binghao Jia, et al.

---

## Author Comment (AC3)

**Response to Anonymous Referee #3:**

Dear Referee #3,

Thank you very much for your helpful comments and suggestions to improve our manuscript. To facilitate this discussion, we first retype your comments in **bold font** and then present our responses in blue to the comments.

**This paper introduces a novel approach to study the effects of surface and subsurface water regulation on dissolved organic carbon (DOC) transport. The authors combine a DOC model with the CLM/MOSART model to simulate the DOC dynamics in both surface water and groundwater systems. This is an innovative contribution because most previous studies and models have neglected the role of water extraction in DOC dynamics and the interaction between DOC and groundwater. However, the paper needs to improve in several aspects before it can be accepted. First, the paper does not clearly quantify the relationship between DOC concentration and water flux in different processes. For instance, how much DOC is removed by surface water and groundwater extraction? Second, the paper does not present the model results in an effective way. Some maps have poorly designed color bars and do not show the spatial patterns clearly. Since the main focus of the paper is on DOC export, some results are not relevant and should be moved to the supplementary information.**

*Response:* We appreciate your clear and detailed feedback, and hope that the response has fully addressed all your concerns. In addition, our model incorporated DOC dynamics schemes to CLM and RTM (River Transport Model) instead of MOSART.

**Page 2**

**Line 30:**

**Rivers are a pipe linking the two major carbon pools of terrestrial and ocean ecosystems.**

**Maybe "aquatic ecosystems"? Since the river also connects to the lake, etc.**

*Response:* Thanks for your comment. Rivers indeed connect other inland aquatic systems such as lakes, but here we want to express that rivers link two major carbon pools: the land and the ocean.

**Line 31:**

**IPCC AR5, full name if it is the first-time usage**

*Response:* We will revise it.

**Line 34:**

[Figure]

**"This", what this is referred to?**

*Response:* This means that the annual DOC transported from terrestrial ecosystems to the ocean via rivers is equivalent to about 1% of the global NPP of terrestrial ecosystems.

**Line 35-37:**

**Higher than what?   Consider breaking this long sentence to shorter ones.**

*Response:* Thanks for your comment. It means riverine DOC is a rather highly reactive organic carbon. And we will revise it as you suggest.

**Line 45**

**May define DOC leaching before use? Also, consider breaking this sentence into shorter ones.**

*Response:* Thanks for your comment, we will revise it.

**Line 61:**

**There are at least some time series DOC measurements in some datasets, even if they are not long-term measurements.**

*Response:* Thank you so much for your careful check. We will revise this sentence to "Moreover, only annual averages are usually available, with no long-term time series variation."

**Line 90:**

**Why not use the same resolution, such as 0.5 degrees, for land and river?**

*Response:* Due to the limitation of available computing resources, the spatial resolution of the model was fixed to $0.9° × 1.25°$ for land, and $0.5° × 0.5°$ for river. Besides, since river module is typically run at a finer resolution than land, the land variables are interpolated to the river grid by coupler in the model.

**Line 103:**

**What do you mean by linear RTM? More details are needed.**

*Response:* CLM5.0 includes two river routing methods: RTM and MOSART. RTM is a linear reservoir method, which uses a linear transport scheme to route water from each grid cell to its downstream neighboring grid cell.

**Line 124:**

**What are upstream and downstream, can you link these processes in Figure 1 so readers can understand what Equation 1 is illustrating in Figure 1?**

*Response:* Please refer to Lines 113-118. The transformation cascade is shown in Fig.S1. And transfers of carbon from upstream to downstream pools in the decomposition cascade are given as:

$$CF_{Lit1,SOM1} = CF_{Lit1}(1 - rf_{Lit1})$$

$$CF_{Lit2,SOM2} = CF_{Lit2}(1 - rf_{Lit2})$$

$$\ldots$$

$$CF_{SOM2,SOM3} = CF_{SOM2}(1 - rf_{SOM2})$$

For more details please see CHAPTER 21 of the CLM5.0 technical notes (Lawrence et al., 2018).

[Figure]

Figure S1:  Schematic of decomposition model (Century model) in CLM. Pool structure, transitions, respired fractions (numbers at end of arrows, rf), and turnover times (numbers in boxes).

**Line 132:**

**The unit of DOC is inconsistent from Line 127. Normally, DOC concentration is expressed as mg C / L. Based on Equation 3, [DOC] is unitless.**

**Again in Line 139, you used another unit for [DOC] as mg g soil-1.**

**Please unify the DOC units.**

**Since you mentioned both adsorption and desorption, you should describe both processes and their equations.**

*Response:* Thanks for your comment.

(1) The unit of [DOC] in our developed model is **g C kgH2O$^{-1}$**, which is calculated by dividing $NS_{DOC}$ (g C m$^{-2}$) by $WS_{tot\_soil}$ (kgH$_2$O m$^{-2}$). In fact, kgH2O is equivalent to L, but in different expressions.

(2) In calculating the sorption coefficient of soil DOC, the concentration unit is normally **mg g soil$^{-1}$**, thus we made a unit conversion during the model calculations.

(3) We have described the adsorption and desorption processes. Please refer to Lines 134-140. In Eq. (5), RE is the amount of DOC desorbed (negative value) or adsorbed (positive value), calculated by the simple initial mass (IM) linear isotherm.

**Line 153:**

**I assume that Equation (7) DOC_leached is the LDOC? If so, you can make them the same.**

**How about RDOC? How is it modeled?**

*Response:* Thanks for your comment. In this study, $R_{DOC}$ is defined as the soil DOC in surface runoff, while $L_{DOC}$ refers to subsurface losses of DOC in soil water. And we will modify it as you suggest.

**Line 188:**

**How are these coefficients obtained? Please provide some details or references.**

*Response:* Thanks for your comment. We will add the reference in our manuscript.

**Besides, how sw and gw are linked to DOC? For example, if there is SW extraction, then shouldn't it be included in Equation (8), such as a term to describe DOC extraction?**

*Response:* Equation (8) is just the riverine DOC transport framework. Please see Sec. 2.5 for more information on how anthropogenic water regulation activities affect DOC transport.

**Line 214:**

**The land component is not 1 degree by 1 degree, so there is some inconsistency.**

*Response:* In the land surface model, 0.9° × 1.25° is usually considered as 1 degree grid resolution.

**Line 220:**

**Is it ok to have spatial interplate dam/reservoir data? Maybe more details are needed to show how it was conducted.**

*Response:* Thanks for your comment, we will add a figure about spatial distribution of reservoir data in the supplementary.

**Line 245:**

**Use scientific notation for large numbers.**

*Response:* We will revise it.

**Line 247:**

**Which was generally consistent? If they are consistent, then it should be "which is consistent". There is no need to use the past sense.**

**There are a few hotspots in the high latitudes, where they are permafrost regions, why are DOC losses relatively higher there?**

*Response:* Thanks for your advice, we will revise this sentence. Besides, due to the large amount of soil organic carbon stored in the permafrost zone. With global warming, the melting of the soil layer in the permafrost zone will be accompanied by the release of organic carbon, especially DOC (Li et al., 2019).

**Line 252:**

**How about a time series evaluation? For example, the DOC concentration at a large river outlet?**

*Response:* Due to the few datasets of long time-series observations of DOC fluxes for large global rivers, we only use the annual datasets to validate the model simulations. In the future, we will collect more observations for time series evaluation.

**Line 254:**

**What do you mean by overestimated or underestimated? Compared with what?**

**In Figure 3d, the color represents both magnitude and over/underestimate. This can be confusing. Since you have a color bar, readers will assume the color represents values based on the color bar.**

*Response:* It is the result of comparing the simulated value with the observed value; if the simulated value is higher than the observed value it is an overestimate and the opposite is an underestimate. In addition, we will add a legend to the figure as you suggest.

**Line 260:**

**This simulation is already a global-scale DOC export study.**

*Response:* Yes, we want to express that our developed model is reasonable.

**Figure 4**

**The positive/negative of latent heat/sensible heat need to be checked. Also, the color bar should match the actual ranges.**

**I am not sure why a multi-year temperature average is needed. If surface water regulation has an impact, then maybe comparing the differences with or without regulation is more meaningful.**

**Soil moisture cannot be negative, so the color bar needs revision.**

**The same applies to runoff.**

*Response:* Figure 4 shows the difference in surface hydrological variables between EXPA and CTL, indicating the effect of surface water regulation, so there exist negative values. In addition, We will revise the title of the figure to make it easier for the reader to understand.

**Figure 7, what is the reference of the change?**

*Response:* In Fig. 7, the blue line is the difference between EXPA and CTL, indicating the effect of surface water regulation; the orange line is the difference between EXPB and EXPA, indicating the effect of groundwater regulation.

**References:**

Lawrence, D., Fisher, R., and Koven, C.: Technical Description of version 5.0 of the Community Land Model (CLM), NCAR, NCAR, Boulder, US, 2018.

Li, M., Peng, C., Zhou, X., Yang, Y., Guo, Y., Shi, G., and Zhu, Q.: Modeling Global Riverine DOC Flux Dynamics From 1951 to 2015, J. Adv. Model. Earth Syst., 11, 514–530, https://doi.org/10.1029/2018MS001363, 2019.

Once again, thank you very much for your comments and suggestions.

Sincerely,

Yanbin You, Zhenghui Xie, Binghao Jia, et al.

---

## Author Response (AR1)

**Point-by-point responses to all review comments**

**NOTE:** To facilitate the evaluation of our responses, original review comments are listed first in their originals (**in bold font**), followed by our itemized responses (in red). An annotated version of revised manuscripts is attached.
* * *
**Referee #1:**

We thank the reviewer for the constructive comments and suggestions, which are in black bold text below. Our itemized response is followed (in red).

1. **Equation (2): Please denote the unit of DOC leaching flux.**

   *Response:* We added the unit of DOC leaching flux (g C m$^{-2}$ s$^{-1}$). Please refer to Line 157.

2. **Line 176: "Riverine DOC is mainly derived from organic carbon leaching processes in soil"; some literature support is required here.**

   *Response:* We added corresponding references as suggested (Gommet et al., 2022; Li et al., 2019). Please refer to Line 136.

3. **Line 189-190: where is the reference for choosing this weighting coefficient?**

   *Response:* We chose this coefficient according to the previous studies (Liu et al., 2019; Zou et al., 2014), and we added corresponding references at the same time. Please refer to Lines 229-230.

4. **Section 3.1: I suggest adding a table to show the main datasets used for model running and validation in this study.**

   *Response:* Thanks for your valuable suggestion. We added a table to summarize the main datasets used in this study. Please refer to Table 1.

5. **Line 221: Please introduce the details for the human water use activity dataset. A description of what data sources were used?**

   *Response:* Based on the comment, we revised the manuscript. Please refer to Lines 260-265.

6. **Line 205: Only the fluxes into the soil carbon pool after surface water extraction are described. What about groundwater extraction?**

   *Response:* Because groundwater extraction usually occurs in situ and will pass through the filtering effect of the soil layer, we hypothesized the part of DOC that returned to soil with groundwater extraction was ignored in our parameterization scheme.

**7. In Section 2.1, the parameters mentioned in the developed soil and river carbon dynamics parameterization scheme are uniform or spatially varying?**

*Response:* The parameters mentioned in our developed schemes are uniform. In fact, it does not correspond to spatial heterogeneity, and we will further refine and modify the parameterization scheme in our future work. The current parameterization scheme has reasonable accuracy in the simulation results, so we believe that our model can be applied to global-scale riverine DOC transport simulation studies.

**8. Line 252: Figures 3a and 3c seem to underestimate. Please check carefully and modify.**

*Response:* Thank you so much for your careful check. We modified it in our manuscript. Please refer to Line 294.

**9. Line 273: Are constants (0.3 and 0.7) in equations the same for the whole world?**

*Response:* Yes, we set the constant due to the limitation of data.

**10. This study developed a model to describe the soil carbon leaching and riverine carbon transport processes, which are not well described in previous land surface models. But the discussion of current uncertainties and limitations in modeling is missing. It should be discussed more.**

*Response:* Thank you for your advice. We added some discussion of current uncertainties and limitations in our manuscript. Please refer to Lines 459-463.

**11. Line 354-355: The authors state that the three rivers were affected by minor groundwater regulation. Please briefly explain the impact and the reasons.**

*Response:* In our selected rivers, only the Mississippi, Yangtze, and Ganges rivers were affected by minor groundwater regulation, which usually occurred during the dry period, where DOC export increased slightly in the Mississippi and Ganges rivers because of higher soil leaching due to irrigation, while DOC export decreased in the Yangtze River due to a significant reduction in river discharge. This also corresponds to the results in Section 4.3.

**12. In section 5, some words about future work are needed.**

*Response:* Thank you for your suggestion. We added some discussion about future work in our manuscript. Please refer to Lines 463-466.

**Referee #2:**

We thank the reviewer for the constructive comments and suggestions, which are in black bold text below. Our itemized response is followed (in red).

**The presentation can be made clearer. The authors should make an effort to make it more accessible to non-modelers (like me), and to readers who want to take home the message without detailed reading of the methods. For example, the different simulations of control conditions and different parts of water regulation considered (CTL, EXPA, EXPB) are in several of the figures presented without explanation or spelling out.**

*Response:* Thank you very much for your suggestion. We modified the subtitle and legend of Fig. 4~12 to make it easier for the reader to understand.

**Is there some way to add (or more carefully discuss) uncertainty ranges around the various estimates and graphs?    The current version presents and compares several numbers with 3-4 significant digits, with no confidence intervals.**

*Response:* Based on your comment, we added a standard deviation after the estimated value to indicate its uncertainty range (mean$\pm$std). Please see Lines 27, 300, 422, 430, and 453.

**Table 2 could be expanded, it appears incomplete. There are several additional estimates of DOC export (possible resulting in a higher median than presented in the manuscript). Some (but not all) are cited in Drake et al. 2018 (Limnol Oceanogr Letters).**

*Response:* Thanks for your detailed comment. We have read the article you mentioned and found that most of the carbon flux which export to the ocean is estimated to be 0.95 Pg C yr$^{-1}$, but it includes all forms of carbon in rivers. We set the organic carbon (OC) / inorganic carbon (IC) ratio to 0.4/0.5, in which 55% of OC flux is dissolved (DOC). Finally, we calculated that the riverine DOC export flux was 232.22 Tg C yr$^{-1}$. We added this result to Table 3 (in the revised manuscript) at the same time. Besides, we also added another result from van Hoek et al. (2021).

**I do not understand how transformations in the regulated and unregulated waters are treated. The methods (line 196) say that "migration transformation" is ignored in the model, and loss rate is assumed equal in reservoirs and rivers. In contrast, one of the model results (line 287) is suggested to be due to increased residence time by the construction of reservoirs, causing increased DOC removal. How is this compatible?**

*Response:* Thank you so much for your careful check. We revised this sentence to "This may be related to the fact that the reservoir adjusting the river discharge and intercepting the riverine DOC." Please refer to Lines 334-335.

[Figure]

**Line 292: "alpine" should be "arctic"**

*Response:* We revised it in our manuscript. Please refer to Line 341.
* * *
[Figure]

**Referee #3:**

We thank the reviewer for the constructive comments and suggestions, which are in black bold text below. Our itemized response is followed (in red).

**Line 30: Rivers are a pipe linking the two major carbon pools of terrestrial and ocean ecosystems. Maybe "aquatic ecosystems"? Since the river also connects to the lake, etc.**

*Response:* Thanks for your comment. Rivers indeed connect other inland aquatic systems such as lakes, but here we want to express that rivers link two major carbon pools: the land and the ocean.

**Line 31: IPCC AR5, full name if it is the first-time usage**

*Response:* We revised it. Please refer to Lines 31-32.

**Line 34: "This", what this is referred to?**

*Response:* This means that the annual DOC transported from terrestrial ecosystems to the ocean via rivers is equivalent to about 1% of the global NPP of terrestrial ecosystems.

**Line 35-37: Higher than what?   Consider breaking this long sentence to shorter ones.**

*Response:* Thanks for your comment. It means riverine DOC is a rather highly reactive organic carbon. And we revised it as you suggest. Please refer to Line 36.

**Line 45: May define DOC leaching before use? Also, consider breaking this sentence into shorter ones.**

*Response:* Thanks for your comment, we revised it. Please refer to Lines 44-47.

**Line 61: There are at least some time series DOC measurements in some datasets, even if they are not long-term measurements.**

*Response:* Thank you so much for your careful check. We revised it. Please refer to Line 73.

**Line 90: Why not use the same resolution, such as 0.5 degrees, for land and river?**

*Response:* Due to the limitation of available computing resources, the spatial resolution of the model was fixed to $0.9° \times 1.25°$ for land, and $0.5° \times 0.5°$ for river. Besides, since river module is typically run at a finer resolution than land, the land variables are interpolated to the river grid by coupler in the model.

**Line 103: What do you mean by linear RTM? More details are needed.**

*Response:* CLM5.0 includes two river routing methods: RTM and MOSART. RTM is a linear reservoir method, which uses a linear transport scheme to route water from each grid cell to its downstream neighboring grid cell. Please refer to Lines 111-116.

**Line 124: What are upstream and downstream, can you link these processes in Figure 1 so readers can understand what Equation 1 is illustrating in Figure 1?**

*Response:* Please refer to Lines 137-142. The transformation cascade is shown in Fig.R1. And transfers of carbon from upstream to downstream pools in the decomposition cascade are given as:

$$CF_{Lit1,SOM1} = CF_{Lit1}(1 - rf_{Lit1})$$

$$CF_{Lit2,SOM2} = CF_{Lit2}(1 - rf_{Lit2})$$

$$\dots$$

$$CF_{SOM2,SOM3} = CF_{SOM2}(1 - rf_{SOM2})$$

For more details please see CHAPTER 21 of the CLM5.0 technical notes (Lawrence et al., 2018).

[Figure]

Figure R1: Schematic of decomposition model (Century model) in CLM. Pool structure, transitions, respired fractions (numbers at end of arrows, rf), and turnover times (numbers in boxes).

**Line 132: The unit of DOC is inconsistent from Line 127. Normally, DOC concentration is expressed as mg C / L. Based on Equation 3, [DOC] is unitless.**

**Again in Line 139, you used another unit for [DOC] as mg g soil-1.**

**Please unify the DOC units.**

**Since you mentioned both adsorption and desorption, you should describe both processes and their equations.**

*Response:* Thanks for your comment.

(1) The unit of [DOC] in our developed model is **g C kgH2O$^{-1}$**, which is calculated by dividing $NS_{DOC}$ (g C m$^{-2}$) by $WS_{tot\_soil}$ (kgH$_2$O m$^{-2}$). In fact, **kgH2O** is equivalent to **L**, but in different expressions.
(2) In calculating the sorption coefficient of soil DOC, the concentration unit is normally **mg g soil$^{-1}$**, thus we made a unit conversion during the model calculations.
(3) We have described the adsorption and desorption processes. Please refer to Lines 163-171. In Eq. (5), RE is the amount of DOC desorbed (negative value) or adsorbed (positive value), calculated by the simple initial mass (IM) linear isotherm.

**Line 153: I assume that Equation (7) DOC_leached is the LDOC? If so, you can make them the same. How about RDOC? How is it modeled?**

*Response:* Thanks for your comment. In this study, the DOC runoff is defined as the soil DOC in surface runoff, and the DOC leaching is defined as the subsurface losses of DOC in soil water. Soil carbon loss ($DOC_{loss}$) is the flux (g C m$^{-2}$ s$^{-1}$), while $R_{DOC}$ and $L_{DOC}$ are the flow (kg C s$^{-1}$), so they use different expressions. We revised it in our manuscript, please refer to Lines 150-162, 177-179.

**Line 188: How are these coefficients obtained? Please provide some details or references.**

*Response:* Thanks for your comment. We added the reference in our manuscript. Please refer to Lines 229-230.

**Besides, how sw and gw are linked to DOC? For example, if there is SW extraction, then shouldn't it be included in Equation (8), such as a term to describe DOC extraction?**

*Response:* Equation (8) is just the riverine DOC transport framework. Please see Sec. 2.5 for more information on how anthropogenic water regulation activities affect DOC transport.

**Line 214: The land component is not 1 degree by 1 degree, so there is some inconsistency.**

*Response:* In the land surface model, 0.9º × 1.25º is usually considered as 1 degree grid resolution.

**Line 220: Is it ok to have spatial interplate dam/reservoir data? Maybe more details are needed to show how it was conducted.**

*Response:* Thanks for your comment, we added a figure about spatial distribution of reservoir data and some details for reservoir operation scheme in the supplement.

**Line 245: Use scientific notation for large numbers.**

*Response:* We revised it. Please refer to Line 287.

**Line 247: Which was generally consistent? If they are consistent, then it should be "which is consistent". There is no need to use the past sense. There are a few hotspots in the high latitudes, where they are permafrost regions, why are DOC losses relatively higher there?**

*Response:* Thanks for your advice, we revised it, please refer to Line 289. Besides, due to the large amount of soil organic carbon stored in the permafrost zone. With global warming, the melting of the soil layer in the permafrost zone will be accompanied by the release of organic carbon, especially DOC (Li et al., 2019).

**Line 252: How about a time series evaluation? For example, the DOC concentration at a large river outlet?**

*Response:* Due to the few datasets of long time-series observations of DOC fluxes for large global rivers, we only use the annual datasets to validate the model simulations. In the future, we will collect more observations for time series evaluation.

**Line 254: What do you mean by overestimated or underestimated? Compared with what? In Figure 3d, the color represents both magnitude and over/underestimate. This can be confusing. Since you have a color bar, readers will assume the color represents values based on the color bar.**

*Response:* It is the result of comparing the simulated value with the observed value; if the simulated value is higher than the observed value it is an overestimate and the opposite is an underestimate. In addition, we added a legend to the figure as you suggest. Please see Fig.3.

**Line 260: This simulation is already a global-scale DOC export study.**

*Response:* Yes, we want to express that our developed model is reasonable.

**Figure 4: The positive/negative of latent heat/sensible heat need to be checked. Also, the color bar should match the actual ranges. I am not sure why a multi-year temperature average is needed. If surface water regulation has an impact, then maybe comparing the**

[Figure]

**differences with or without regulation is more meaningful. Soil moisture cannot be negative, so the color bar needs revision.   The same applies to runoff.**

*Response:* Figure 4 shows the difference in surface hydrological variables between EXPA and CTL, indicating the effect of surface water regulation, so there exist negative values. In addition, we revised the title of the figure to make it easier for the reader to understand.

**Figure 7, what is the reference of the change?**

*Response:* In Fig. 7, the blue line is the difference between EXPA and CTL, indicating the effect of surface water regulation; the orange line is the difference between EXPB and EXPA, indicating the effect of groundwater regulation.

[Figure]

**References**

Gommet, C., Lauerwald, R., Ciais, P., Guenet, B., Zhang, H., and Regnier, P.: Spatiotemporal patterns and drivers of terrestrial dissolved organic carbon (DOC) leaching into the European river network, Earth Syst. Dynam., 13, 393–418, https://doi.org/10.5194/esd-13-393-2022, 2022.

van Hoek, W. J., Wang, J., Vilmin, L., Beusen, A. H. W., Mogollón, J. M., Müller, G., Pika, P. A., Liu, X., Langeveld, J. J., Bouwman, A. F., and Middelburg, J. J.: Exploring Spatially Explicit Changes in Carbon Budgets of Global River Basins during the 20th Century, Environ. Sci. Technol., 55, 16757–16769, https://doi.org/10.1021/acs.est.1c04605, 2021.

Lawrence, D., Fisher, R., and Koven, C.: Technical Description of version 5.0 of the Community Land Model (CLM), NCAR, NCAR, Boulder, US, 2018.

Li, M., Peng, C., Zhou, X., Yang, Y., Guo, Y., Shi, G., and Zhu, Q.: Modeling Global Riverine DOC Flux Dynamics From 1951 to 2015, J. Adv. Model. Earth Syst., 11, 514–530, https://doi.org/10.1029/2018MS001363, 2019.

Liu, S., Xie, Z., Zeng, Y., Liu, B., Li, R., Wang, Y., Wang, L., Qin, P., Jia, B., and Xie, J.: Effects of anthropogenic nitrogen discharge on dissolved inorganic nitrogen transport in global rivers, Glob Change Biol, 25, 1493–1513, https://doi.org/10.1111/gcb.14570, 2019.

Zou, J., Xie, Z., Yu, Y., Zhan, C., and Sun, Q.: Climatic responses to anthropogenic groundwater exploitation: a case study of the Haihe River Basin, Northern China, Clim Dyn, 42, 2125–2145, https://doi.org/10.1007/s00382-013-1995-2, 2014.

---

## Author Response (AR2)

**Point-by-point responses to all review comments**

__NOTE:__ To facilitate the evaluation of our responses, original review comments are listed first in their originals (**in bold font**), followed by our itemized responses (in red).
* * *
**Editor's Comments:**

**The reviewer report(s) for your manuscript are now complete. According to the reports, they recommend that you make some revisions before further consideration. Please consider the reviewers' comments and amend your manuscript according to their recommendations.**

*Response:* Thank you for your information and the two anonymous reviewers' comments. Those comments are very helpful for revising and improving our paper. We have revised the manuscript based on their comments and suggestions.

**Referee #3:**

We thank the reviewer for the constructive comments and suggestions, which are in black bold text below. Our itemized response is followed (in red).

**The manuscript still has various issues that require clarification. More details of the modeling parts are needed.**

*Response:* Based on your comment, we added more details of the modeling part.

(1) To facilitate understanding, we modified the descriptions about the DOC runoff and leaching flux, please see Lines 129-135:

"*The fluxes are described as follows:*

$$DOC_{runoff} = [DOC]Q_{surf}k_{adsorb} - SR, \qquad (2)$$

$$DOC_{leaching} = [DOC]Q_{dis}k_{adsorb} - SR, \qquad (3)$$

*where $DOC_{runoff}$ (g C m$^{-2}$ s$^{-1}$) denotes the soil DOC runoff, $DOC_{leaching}$ (g C m$^{-2}$ s$^{-1}$) denotes the soil DOC leaching, $Q_{surf}$ (kgH$_2$O m$^{-2}$ s$^{-1}$) denotes the surface runoff, $Q_{dis}$ (kgH$_2$O m$^{-2}$ s$^{-1}$) denotes the subsurface discharge*".

(2) We added Equation (11) for the effective riverine flow velocity, please see Lines 170-172: "

$$v = max(0.05, \beta^{1/2}), \tag{11}$$

*where $v$ (m s$^{-1}$) is the effective riverine flow velocity, which is estimated by grid cell mean topographic slope $\beta$ (Oleson et al., 2013); ".*

(3) In addition, we added some detailed statements on the processes of DOC transfer induced by reservoir interception, surface water withdrawal, and groundwater extraction, please see Lines 213-225:

" *The process of reservoir interception leading to retention of carbon in rivers can be expressed as:*

$$F_{DOC,r} = \frac{v(con_r \Delta Q_r)}{d}, \tag{18}$$

*where $F_{DOC,r}$ (kg C s$^{-1}$) denotes the DOC flux retained by the reservoir; $con_r$ (kg C m$^{-3}$) is the DOC concentration in the reservoir; $\Delta Q_r$ (m$^3$) is the water volume change in the reservoir. Therefore, the riverine DOC flux leaving the current grid cell is updated to:*

$$F_{DOC}^{out} = F_{DOC}^{out} - F_{DOC,r}, \tag{19}$$

*The DOC flux extracted from surface water is calculated based on the intake rate and the solute concentration $con_{DOC}$ (kg C m$^{-3}$) in the current grid cell and return to the soil DOC pool after irrigation:*

$$F_{DOC}^{out} = F_{DOC}^{out} - q_{sw} con_{DOC}, \tag{20}$$

*The reduction in soil DOC leaching due to groundwater extraction is then calculated based on soil DOC concentration and groundwater pumping rate:*

$$DOC_{leaching} = DOC_{leaching} - q_{gw}[DOC]. \tag{21}$$

".

For more details please see Sect.2 in the revised manuscript.

We also added thanks to the editor and two anonymous reviewers for the helpful comments in the acknowledgments section.